# Targeting mTOR and Glycolysis in HER2-Positive Breast Cancer

**DOI:** 10.3390/cancers13122922

**Published:** 2021-06-11

**Authors:** Ryan W. Holloway, Paola A. Marignani

**Affiliations:** Department of Biochemistry & Molecular Biology, Faculty of Medicine, Dalhousie University, Halifax, NS B3H 4R2, Canada; rhollowa@dal.ca

**Keywords:** mTOR, glycolysis, HER2-positive breast cancer, 2-deoxyglucose, trastuzumab, Herceptin, LKB1, tumor recurrence, rapalog, PI3K-Akt-mTOR signaling pathway, clinical diagnostics

## Abstract

**Simple Summary:**

About one third of all breast cancers are classified as HER2-positive due to high levels of the HER2 cell surface protein. Drugs that target HER2 have been mostly successful, but this type of cancer returns at a high frequency once treatment has been completed. The high levels of HER2 also cause elevated activation of mechanistic target of rapamycin (mTOR) and enhanced glucose metabolism, both of which support cancer growth. Based on this, drugs have been developed to block mTOR and tested in clinical studies alone or in combination with drugs that target HER2. These treatments are successful but have more toxic effects and a higher chance that the cancer will return. Using drugs that mimic glucose deprivation in HER2-positive breast cancer patients has not been tested; however, preclinical studies have shown HER2-positive breast tumors are reduced by combining drugs that mimic glucose deprivation with mTOR inhibitors.

**Abstract:**

Up to one third of all breast cancers are classified as the aggressive HER2-positive subtype, which is associated with a higher risk of recurrence compared to HER2-negative breast cancers. The HER2 hyperactivity associated with this subtype drives tumor growth by up-regulation of mechanistic target of rapamycin (mTOR) pathway activity and a metabolic shift to glycolysis. Although inhibitors targeting the HER2 receptor have been successful in treating HER2-positive breast cancer, anti-HER2 therapy is associated with a high risk of recurrence and drug resistance due to stimulation of the PI3K-Akt-mTOR signaling pathway and glycolysis. Combination therapies against HER2 with inhibition of mTOR improve clinical outcomes compared to HER2 inhibition alone. Here, we review the role of the HER2 receptor, mTOR pathway, and glycolysis in HER2-positive breast cancer, along with signaling mechanisms and the efficacy of treatment strategies of HER2-positive breast cancer.

## 1. Introduction

In women, the most diagnosed cancer and a leading cause of cancer-related deaths worldwide is breast cancer [1,2]. The highest incidence of breast cancer occurs in the United States and Western Europe, whereas the lowest incidence is found in Africa and Asia. Black American women have the highest mortality rates of breast cancer, and Korean women have the lowest mortality rates [2]. Although mortality rates due to breast cancer are declining in developed regions of the world, mortality rates are increasing in developing regions where limitations for early detection and awareness of symptoms are contributing factors [3,4,5,6]. The type of breast cancer is characterized by overexpression or absence of hormonal receptors: estrogen receptor-positive (ER+), progesterone-positive (PR+), human epidermal growth factor receptor 2-positive (HER2-positive), or the absence of ER/PR/HER2 (triple-negative). HER2-positive breast cancer constitutes 15–30% of all breast tumors [7,8,9]. Based on the gene expression-based test PAM50 [10], HER2-positive breast cancers have been grouped into four intrinsic molecular subtypes: luminal A, luminal B, HER2-overexpressing or HER2-enriched (HER2-E), and basal-like. Of these subtypes, HER2-E constitutes about 50–60% of all HER2-positive breast cancers [11,12,13]. HER2-positive breast tumors progress faster and more aggressively than most other breast tumors. Anti-HER2 therapies often lead to the development of chemoresistance (reviewed in [14]) and an elevated risk of recurrence that increases mortality rates [15] (reviewed in [16]). HER2-positive breast cancer is associated with hyperactivation of the mTOR pathway and a metabolic shift from aerobic respiration to glycolysis (Figure 1). As the mechanistic target of rapamycin (mTOR) pathway [17] and glycolysis [18] also contribute to supporting tumor recurrence and chemoresistance, these signaling pathways have become appealing targets for HER2-positive breast cancer therapy.

## 2. Neu/HER2 in Breast Cancer

### 2.1. Neu/HER2 Biology

HER2 is a member of the epidermal growth factor (EGF) receptor family comprising EGFR/ErbB1, HER2/ErbB2, HER3/ErbB3, and HER4/ErbB4. EGF receptor family proteins encode for transmembrane receptors consisting of the extracellular ligand binding, lipophilic transmembrane, and cytoplasmic tyrosine kinase domains. Ligand binding to EGF receptors activates their hetero- or homodimerization whereby one EGF receptor will phosphorylate the other on a tyrosine, allowing cytoplasmic signaling complexes to bind to the EGFR dimer. This stimulates signal transduction cascades such as the PI3K-Akt-mTOR and MAPK pro-survival pathways [14,19] (reviewed in [20]). However, the proto-oncogene Neu/HER2 (HER2) has no known natural ligands [21]; thus, to elicit downstream signaling, HER2 preferentially dimerizes with EGF receptors that subsequently results in a more potent signaling response. Due to its role in activating signaling pathways that stimulate cell proliferation and survival, HER2 overexpression in breast tumor cells promotes tumor growth and increases the percentage of cells in S phase, aneuploidy, and migration of cancer cells into the lymph nodes [22] (reviewed in [23]). Spontaneous homodimerization of HER2 is more prominent in HER2-positive breast cancer, shifting the balance of the EGR receptor dimers towards a greater ratio of homodimers [19,24]. Following dimerization, EGF receptors undergo endocytosis and are either recycled back to the cell surface or targeted for proteasomal degradation by ubiquitin ligase Cbl [25]. Dimerization stimulates EGFR/ErbB1 phosphorylation (Y1045) and recruitment of Cbl [25,26]; however, in heterodimers with ErbB2, Cbl recruitment is impeded. Although ErbB2 is phosphorylated (Y1112), Cbl recruitment is inefficient, and EGF receptors are recycled back to the cell surface [27].

### 2.2. HER2 Inhibitors: Mechanisms of Action and Efficacy in Clinical Studies

#### 2.2.1. Trastuzumab (Herceptin^®^)

The humanized monoclonal antibody trastuzumab (Herceptin^®^) was the first clinically used therapeutic agent to target HER2 for HER2-positive breast cancer (Figure 1) [28]. Trastuzumab binds to subdomain IV of the extracellular domain (ECD) of HER2; however, this does not inhibit HER2 dimerization or tyrosine kinase activity [29]. Rather, trastuzumab binding to HER2 induces antibody-dependent cellular cytotoxicity (ADCC) in which HER2-bound trastuzumab directs natural killer (NK) cells to destroy tumor cells [30]. Trastuzumab induction of ADCC in HER2-positive breast cancer cells was observed in allograph mouse models using tumor cells of Neu/HER2-expressing mice [30] and later observed in clinical trials [27,31,32]. Furthermore, HER2-specific monoclonal antibodies directed towards several epitopes were effective at reducing tumor growth in vivo [31] and enhanced the recruitment of Cbl to phosphorylated HER2 (Y1112), followed by its proteasomal degradation [31]. Phase II studies using trastuzumab in combination with other therapies showed a higher overall response rate (ORR) and median progression-free survival (PFS) than trastuzumab monotherapy (Table 1) [32,33,34]. Despite the success of trastuzumab therapy, recurrence-free survival is low (11–18%) during the first 5 years of remission [35]. This relapse is common and frequently associated with brain metastases due to the inability of antibody-based treatments, such as trastuzumab, to traverse the blood–brain barrier [36]. Trastuzumab has been associated with injury to cardiac myocytes, although this effect was determined to be reversible. While the preliminary phase II trials observed minimal adverse cardiac effects [32,33], a phase III trial using patients, of whom many had previously received anthracycline therapy, had a high rate of cardiac dysfunction [37]. Evidence later indicated that the adverse cardiac effects were fairly low for trastuzumab monotherapy and prior anthracycline treatment (3–7%) and anthracycline plus cyclophosphamide (8%), but trastuzumab and anthracycline plus cyclophosphamide treatment showed a much higher incidence (27%) [38]. HER2 activity was also found to play an essential role in regulating cardiac function and to prevent anthracycline cardiotoxicity [39,40].

#### 2.2.2. Lapatinib

Lapatinib (GW2016) is a synthetic small-molecule reversible inhibitor of EGFR and HER2 tyrosine kinase activity by acting as an ATP-competitive inhibitor. Lapatinib selectively inhibits the kinase activity of EGFR and HER2 and also prevents their autophosphorylation [60]. Konecny et al. [61] evaluated the therapeutic ability of lapatinib using a panel of 31 breast cancer cell lines that also included trastuzumab-resistant HER2-positive breast cancer cells. Lapatinib treatment of HER2-positive breast cancer cell lines was shown to inhibit HER2 and EGFR activity as well as downstream phosphorylation of Akt and ERK. Xenograft mouse models implanted with HER2-positive breast cancer cells showed a significant reduction in tumor volume in the lapatinib treatment group compared to the control group. Furthermore, lapatinib inhibited the growth of trastuzumab-resistant breast cancer cell lines in vitro. Phase II and III clinical trials using lapatinib treatment demonstrated clinical benefit in nearly one third of the patients with advanced HER2-positive breast tumors (Table 1). Although lapatinib was well-tolerated, low-grade adverse effects were frequent [41,43,44,45]. Despite the clinical benefit of lapatinib, resistance develops through re-activation of mTOR signaling (Figure 1) and up-regulation of nuclear receptor ERRα, a key regulator of cell metabolism that is normally degraded in response to lapatinib treatment [62].

#### 2.2.3. Neratinib

Neratinib (or HKI-272) inhibits HER2 tyrosine kinase activity by covalent bonding to the cysteine residue of the ATP pocket. In contrast to lapatinib, neratinib is an irreversible inhibitor of HER1, HER2, and HER4, in which it inhibits EGF receptor dimerization and kinase activity [63,64]. Using a panel of 36 breast cancer cell lines, Canonici et al. [64] showed that neratinib was more potent than lapatinib in inhibiting HER2 activity and downstream phosphorylation of Akt and ERK in HER2-positive cell lines, including those that have acquired trastuzumab resistance. Rabindran et al. [63] examined the effects of neratinib using human HER2-positive breast cancer cell lines, where it was found neratinib treatment down-regulated MAPK, Akt, and RB phosphorylation, down-regulated cyclin D1 expression, and up-regulated the cell cycle inhibitor p27 in a dose-dependent manner. Furthermore, preclinical experiments showed that neratinib treatment reduced tumor growth in xenograft mouse models implanted with HER2-positive breast cancer cells compared to the controls [63,64]. Furthermore, the combination of neratinib and trastuzumab was more effective at reducing tumor growth than the monotherapies [64]. Phase II trials observed high response rates and median PFS in neratinib treatment of HER2-positive breast cancer (Table 1), but these were lower in patients with prior trastuzumab treatment. Like lapatinib, neratinib was well-tolerated as a monotherapy [47], and resistance to neratinib treatment was also attributed to the hyperactivation of the mTOR signaling pathway (Figure 1) [65]. In the phase III NALA trial, patients previously treated for metastatic HER2-positive breast cancer responded more favorably to neratinib plus capecitabine, particularly HER2-positive breast cancer patients that were hormone receptor-negative (HR−) [48]. However, the exteNET trial found that hormone receptor-positive (HR+) patients responded more favorably to neratinib treatment compared to HR− patients [66].

#### 2.2.4. Pertuzumab (2C4)

Like trastuzumab, pertuzumab (2C4) is a HER2 humanized monoclonal antibody that induces ADCC [67], but pertuzumab binds to a different epitope of the ECD of HER2 than trastuzumab. Pertuzumab inhibits HER2 dimerization with HER3, unlike trastuzumab which inhibits HER2 homodimerization [68]. In preclinical experiments, Scheuer et al. [69] compared the efficacy of pertuzumab, trastuzumab, and both drugs in combination in treating xenograft mouse models of cancer. They found that monotherapy of pertuzumab reduced the tumor volume to a similar extent as the trastuzumab treatment, but tumor growth was significantly quenched with dual therapy of both monoclonal antibodies. Phase II studies found pertuzumab/trastuzumab combined therapy was successful in treating HER2-positive breast cancer (Table 1), with more than 10% of patients experiencing adverse effects, but the therapy was otherwise well-tolerated [49,50,51]. In the phase III Cleopatra (CLinical Evaluation Of Pertuzumab And TRAstuzumab) study, Baselga et al. [50] treated 808 patients with pertuzumab plus trastuzumab plus docetaxel (an inhibitor of tubulin assembly/disassembly) or trastuzumab plus docetaxel as the control group. Here, the addition of pertuzumab significantly increased PFS by several months without any increase in cardiac toxicity (Table 1). In the APHINITY clinical trials [52,70], patients with early HER2-positive breast cancer were treated with pertuzumab or placebo in combination with trastuzumab and chemotherapy. HER2-positive breast cancer patients treated with pertuzumab showed significant improvement in the 3-year invasive disease-free survival rate compared to the placebo group. Furthermore, disease recurrence in the pertuzumab group was reduced. Much like the phase II studies, the addition of pertuzumab to a trastuzumab and chemotherapy treatment regimen did not result in any significant increase in adverse effects other than elevated diarrhea scores [70].

#### 2.2.5. Trastuzumab Antibody–Drug Conjugates

Trastuzumab emtansine (trastuzumab-DM1) is an antibody–drug conjugate where trastuzumab is stably bound to DM1, a derivative of anti-tumor drug maytansine. Trastuzumab-DM1 binds to HER2 and is subsequently internalized, where DM1 is released and inhibits microtubule assembly/disassembly and cell proliferation. In phase II clinical trials, trastuzumab-DM1 monotherapy showed similar success compared to other HER2 therapies in HER2-positive breast cancer (Table 1) and was well-tolerated with only ~20% of patients experiencing adverse effects [53]. Another trastuzumab antibody–drug conjugate is trastuzumab deruxtecan (DS-8201), which is trastuzumab linked to a cytotoxic topoisomerase I inhibitor by a tetrapeptide-based linker that is selectively cleaved by cathepsins, which are up-regulated in the tumor cells. In contrast to trastuzumab-DM1, DS-8201 releases a drug with improved cell membrane permeability and has a shorter half-life [71,72]. In phase II clinical trials, DS-8201 showed an ORR of 60.2% in patients with HER2-positive metastatic breast cancer [55]. Anti-tumor effects were even observed in patients with low HER2 expression. Although gastrointestinal and hematologic toxicity was a common occurrence in patients during treatment, no cardiotoxicity was observed, but an increased risk of interstitial lung disease was associated with treatment.

#### 2.2.6. Tucatinib

Like lapatinib, tucatinib binds to the ATP pocket of HER2 and acts as a competitive, reversible tyrosine kinase inhibitor, but tucatinib is selective for HER2 only [73]. Kulukian et al. [73] also determined that tucatinib effectively inhibited HER2 phosphorylation in vitro using the HER2-positive breast cancer cell line BT-474 and observed a minimal inhibition of EGFR phosphorylation in the EGFR-overexpressing skin cancer cell line A431. In xenograft mouse models of HER2-positive breast cancer implanted with BT-474 cells, tucatinib treatment resulted in a delay in tumor growth comparable to trastuzumab monotherapy, and these effects were enhanced in the combination therapy of tucatinib plus trastuzumab [73]. In HER2CLIMB, a phase II clinical trial, tucatinib–trastuzumab–capecitabine combination therapy for HER2-positive breast cancer showed a higher ORR and median PFS than the placebo–trastuzumab–capecitabine group (Table 1); however, this was associated with a high incidence of adverse effects [56].

#### 2.2.7. Pyrotinib

Pyrotinib (or SHR1258) is an irreversible inhibitor of EGFR/HER1, HER2, and HER4 shown to suppress tumor growth in HER2-positive breast cancer xenograft mouse models, which was associated with a favorable safety profile [74]. In phase I and II clinical trials, Ma et al. [57,58] reported that the combination therapy of pyrotinib and capecitabine resulted in significantly improved ORR and PFS as compared to lapatinib and capecitabine therapy (Table 1). Pyrotinib treatment was also well-tolerated, with a low percentage of patients experiencing grade 3 adverse events.

#### 2.2.8. HLX02

Although success using trastuzumab to treat HER2-positive breast cancer has been achieved, the expensiveness of this drug limits accessibility to patients. Hence, biosimilar drugs ameliorate this issue as they are nearly indistinguishable from the original drug and are less expensive. Recently, a biosimilar to trastuzumab, HLX02 [75], was shown to have near-identical clinical results when compared to trastuzumab [76]. Xu et al. [59] reported that HXL02 showed similar efficacy and adverse effects in patients with HER2-positive recurrent or metastatic breast cancer compared to patients receiving trastuzumab (Table 1).

## 3. The mTOR Pathway in HER2-Positive Breast Cancer

### 3.1. mTOR Protein Structure and Function

Enhanced activity of mTOR is associated with HER2-overexpressing breast cancers [77,78]. The mTOR protein contains five domains: (i) the Huntington, EF3A, ATM, TOR (HEAT) repeats, (ii) the helical Frap, ATM, TRRAP (FAT) domain, (iii) FKBP–rapamycin complex binding (FRB), (iv) the kinase domain, and (v) FAT C-terminal (FATC). mTOR is the catalytic subunit in two protein complexes, mTOR complex 1 (mTORC1) and mTOR complex 2 (mTORC2). mTORC1 is composed of five components: mTOR, regulatory-associated protein of mTOR (raptor), mammalian lethal with Sec13 protein 8 (mLST8 or GβL), proline-rich Akt substrate 40 kDa (PRAS40), and DEP-domain-containing mTOR-interacting protein (deptor). Like mTORC1, mTORC2 also contains mTOR, mLST8, and deptor, but has the rapamycin-insensitive companion of mTOR (rictor), mammalian stress-activated protein kinase interacting protein (mSIN1), and protein observed with Rictor-1 (Protor-1) subunits [79]. Raptor mediates mTORC1 binding to the substrates for phosphorylation [80]. Rictor is unrelated to raptor but facilitates the phosphorylation of various substrates of mTORC2 [81].

### 3.2. Regulation of mTOR Pathway

Serine-threonine kinase LKB1 (STK11, also referred to as Liver Kinase 1), is a tumor suppressor that negatively regulates mTORC1 activity. LKB1 functions within a heterotrimer, containing the pseudokinase STRAD and the adaptor protein MO25 [82,83,84], that activates AMPK by phosphorylation on T172 (Figure 1; [85]). AMPK, also a serine-threonine kinase, is a metabolic sensor and negative regulator of the mTOR signaling pathways. The AMPK protein is composed of the catalytic α subunit and the regulatory β and γ subunits. AMPK activation is induced by energy-stressed conditions in which intracellular AMP levels are elevated. AMP activates AMPK by binding to the γ subunit and subsequently targeting it for phosphorylation by LKB1 on T172 [86]. AMPK activates the tuberous sclerosis complex 1 and 2 (TSC1/2) by phosphorylation of T1227 and S1345 on TSC2 [87]. Activated TSC1/2 acts as a GTPase activating protein (GAP) that inhibits the small GTPase Ras Homolog Enriched in Brain (Rheb), which is involved in mTORC1 activation [88]. In addition, AMPK can directly inhibit mTORC1 by phosphorylation of raptor on S722 and S792 [89].

The PI3K-Akt pathway is an important regulator of the mTOR pathway. The Akt protein consists of three main domains: the N-terminal pleckstrin homology (PH) domain, the central kinase catalytic (CAT) domain, and the C-terminal extension (EXT) region. The PH domain mediates the protein–protein and protein–lipid interactions of Akt. The CAT domain mediates enzymatic activity and contains T308, a phosphoinositide-dependent kinase 1 (PDK1)-dependent phosphorylation site necessary for Akt activation [90]. The EXT region contains S473 phosphorylation for full Akt activation (reviewed in [91]). Akt enables mTOR activity by inactivating TSC1/2 through the phosphorylation of TSC2 on several residues, hence enabling Rheb-GTPase to activate mTORC1 (Figure 1; [92]). Akt can also directly activate mTORC1 by phosphorylation of mTOR on S2448 [93].

The cyclic adenosine 3′,5′-monophosphate (cAMP)-dependent protein kinase A (PKA) is a key regulator of the mTOR pathway, and both are up-regulated in metastatic breast cancer [78]. PKA consists of two catalytic subunits and two regulatory subunits and can be activated under glucose-deprived conditions where an accumulation of cellular levels of cAMP occurs. Activation of PKA by cAMP occurs when cAMP binds to the regulatory subunits of PKA, which dissociates them from the catalytic subunits that are responsible for PKA activities [94,95,96]. Once activated, PKA promotes the phosphorylation and activation of AMPK in an LKB1-dependent manner [97], which then leads to the inhibition of mTORC1 by phosphorylation of raptor on S791 [98]. The study by Kari et al. [97] found that the knockout of LKB1 significantly reduced AMPK phosphorylation in response to PKA activation; however, the reduction in LKB1 did not completely ablate this effect, possibly due to residual LKB1. They also speculated that AMPK phosphorylation is mediated by another kinase. Other than LKB1, Ca^2+^/calmodulin-dependent protein kinase kinase 2 (CaMKK2) [99] and transforming growth factor-β (TGF-β)-activated kinase 1 (TAK1) [100,101] are both known to phosphorylate AMPK on T172. Since this review is focused on discussing the role of AMPK in regulating the mTOR pathway, further discussion regarding the regulation of AMPK has been reviewed in detail elsewhere [102].

### 3.3. Downstream of mTOR Pathway

#### 3.3.1. mTORC1 Activation of S6K1 and 4E-BP1

mTORC1 activates translational regulators p70 ribosomal S6 kinase 1 (S6K1) and eukaryotic initiation factor 4E (eIF4E)-binding protein 1 (4E-BP1) (Figure 1). 4E-BP1 acts as a translational inhibitor by binding to eIF4, thus inhibiting eIF4 from binding with eIF4G. This prevents incorporation of eIF4E into the eIF4F complex, consequently halting 5’-cap-dependent translation [103]. mTORC1 phosphorylates 4E-BP1 on T37 and T46, which primes 4E-BP1 for subsequent phosphorylation that dissociates it from eIF4E, thereby enabling 5′-cap-dependent translation [104]. Activated mTORC1 also mediates S6K1 activity by binding to the eIF3 complex with S6K1. mTORC1 then phosphorylates S6K1, leading to its dissociating from the eIF3 complex, which subsequently allows S6K1 activity [105].

#### 3.3.2. mTORC1 Regulation of Cell Cycle

Activation of mTORC1 also stimulates cell cycle progression through S6K1 and 4E-BP1. Fingar et al. [106] showed that inhibition of mTORC1 activity and knockdown of S6K1 and 4E-BP1 all resulted in the inhibition of G1 phase progression in U2OS osteosarcoma cells. Averous et al. [107] showed that mTORC1 regulated the expression of cyclin D1, an essential regulator of the G1/S phase transition, in a 4E-BP1-dependent manner. Inhibition of mTORC1 activity down-regulated cyclin D1 mRNA and protein levels in MCF7 breast cancer cells, but this effect was rescued when eIF4E activity was enhanced by the knockdown of 4E-BP1. Enhanced activity of eIF4E also increased cyclin D1 translation as knockdown of 4E-BP1 resulted in an increased association of polysomes to the cyclin D1 mRNA. Furthermore, overexpression of 4E-BP1, in the absence of active mTORC1, led to decreased cyclin D1 levels. Knudson et al. [108] observed cyclins A, D1, and E1 were all down-regulated in response to dual inhibition of mTORC1 and mTORC2, knockdown of raptor, and dual knockdown of raptor and rictor.

#### 3.3.3. mTORC1 Regulation of Akt

Although Akt is a positive regulator of mTORC1 activation, mTORC1 negatively regulates Akt activation by modulating its activation by insulin receptor substrate-1 (IRS-1). mTORC1-activated S6K1 inactivates IRS-1 by phosphorylation on S422 [109], leading to its proteasomal degradation [110]. The mTORC1-dependent activation of S6K1 negatively regulates the ERK/MAPK pathway. Inhibition of mTORC1 up-regulated phosphorylated ERK (T202/Y204) levels, which was ablated by the overexpression of constitutively active S6K1 [111].

#### 3.3.4. mTORC1 Regulation of Glycolysis

The mTOR pathway has been implicated as a key regulator of metabolic functions. mTORC1 is involved in glucose uptake and glycolysis by up-regulating the activation of transcription factors such as HIF1α. Using murine embryonic fibroblasts (MEFs) from wild-type, *Tsc1*^−/−^, and *Tsc2*^−/−^ mice, Duvel et al. [112] demonstrated that activated mTORC1 caused an elevated expression of HIF1α in *Tsc2*^−/−^ MEFs, but not in *Rictor*^−/−^ MEFs. The mTORC1-dependent up-regulation of HIF1α was shown to be dependent on 4E-BP1 and not S6K1. Knockdown of raptor or inhibition of mTORC1 activity both down-regulated transcript levels of glycolytic enzymes *Glut1*, *Pfkp*, and *Pdk1*, which was consistent with the knockdown of HIF1α. Increased glucose uptake due to mTOR hyperactivation was observed in *Tsc2*^−/−^ MEFs, and this effect was blocked by knockdown of HIF1 and inhibition of mTORC1 activity, thus indicating that mTORC1 mediated the increase in glucose uptake through HIF1α. Finlay et al. [113] also demonstrated that activation of mTORC1, independent of the PI3K-Akt pathway, was found to up-regulate the expression of HIF1α in CD8^+^ T cells. Poulain et al. [114] examined the effect of mTORC1 activation on glycolytic activity in acute myeloid leukemia (AML) cells. Hyperactive mTORC1 in AML cells was observed to enhance glycolysis and establish a dependency on glycolysis for survival. Furthermore, transcriptome analysis of AML cell line MOLM-14 treated with an inhibitor of mTORC1 showed a down-regulation of genes involved in glycolysis and PPP. Furthermore, inhibition of mTORC1 also reduced glucose uptake and lactate production in four different AML cells with hyperactive mTOR signaling. In addition to its role in glucose metabolism, mTORC1 also regulates lipid/cholesterol metabolism. mTORC1 was shown to mediate the activity of two transcription factors that regulate lipid and cholesterol homeostasis, sterol regulatory element-binding protein (SREBP) [112,115], and peroxisome proliferator-activated receptor-γ (PPARγ) [116].

#### 3.3.5. mTORC1 Regulation of Mitochondrial Biogenesis

mTORC1 was found to be essential for mitochondrial biogenesis and activity in a 4E-BP1-dependent manner in vitro and in vivo. Inhibition of mTORC1 or raptor knockdown both down-regulated transcription of genes involved in oxidative phosphorylation and genes encoding mitochondrial ribosomal proteins and reduced mitochondrial respiration, intracellular ATP levels, and mitochondrial DNA content [117]. Cunningham et al. [118] observed that the inhibition of mTORC1 resulted in the down-regulation of several genes involved in mitochondrial function and reduced mitochondrial respiration. mTORC1 also interacts with the transcriptional regulators yin-yang 1 (YY1) and peroxisome-proliferator activated receptor coactivator-1a (PGC-1a), involved in mitochondrial biogenesis and oxidative metabolism. Consistent with these findings, the loss of mTORC1 activity in the skeletal muscle from conditional knockout of raptor in mice also showed a reduction in transcript levels of PGC-1a and its target gene myoglobin, protein levels of the PGC-1a co-activator PPARγ and the mitochondrial marker cytochrome *c* oxidase IV (COX IV), and oxygen capacity [119]. Morita et al. [117] demonstrated the involvement of mTORC1 in regulating mitochondrial biogenesis and activity in a 4E-BP1-dependent manner in vitro and in vivo. They found that inhibition of mTORC1 modulated the mRNA levels of several genes comprising the components of complex V of the oxidative phosphorylation pathway, TFAM (a regulator of mitochondrial DNA replication and transcription), numerous mitochondrial ribosomal proteins (MRPLs), and NADH dehydrogenase 1 alpha subcomplex assembly factors 2 and 4 (NDUFAF2 and 4). The knockdown of mTORC1 subunit raptor resulted in a reduction in ATP synthase subunit ATP5O and TFAM expression, mitochondrial respiration, TCA intermediates pyruvate and lactate, and intracellular ATP levels and mitochondrial DNA content as compared to the control.

#### 3.3.6. mTORC1 Regulation of Autophagy

mTORC1 is a positive regulator of cell growth by repressing autophagy through inhibition of unc-51-like kinase 1 (ULK1) [120], a kinase that initiates autophagy by promoting autophagosome formation. ULK1 is present as a complex with autophagy-related gene-13 (ATG13), autophagy-related gene-101 (ATG101), and focal adhesion kinase family-interacting protein of 200 kDa (FIP200). In glucose-starved conditions, ULK1 interacts with AMPK and is subsequently activated by phosphorylation on S317 and S777. In nutrient-sufficient conditions, mTORC1 phosphorylates ULK1 on S757, which prevents autophagosome formation by disrupting the interaction of ULK1 with AMPK [121].

#### 3.3.7. mTORC1 Regulation of mTORC2

The activation of mTORC1 maintains a feedback loop that inhibits mTORC2 activity. This negative feedback loop is initiated by mTORC1 phosphorylation of S6K1 that subsequently phosphorylates the mTORC2 subunits rictor on T1153 and mSin1 on Thr86 and Thr398 [122,123,124]. The phosphorylation of rictor did not affect mTORC2 assembly, kinase activity, or cellular localization; however, mutation of T1153 resulted in increased mTORC2 activity [122]. In contrast, the phosphorylation of mSin1 causes the dissociation of mSin1 from mTORC2, thus preventing mTORC2 activity [124]. AMPK activation can affect the negative feedback loop by inhibiting mTORC1 and alleviating the inhibition of mTORC2 [125]. Gao et al. [125] showed that the non-steroidal anti-inflammatory drug aspirin induced the activation of AMPK, leading to the up-regulation of mTORC2-dependent phosphorylation of Akt (S473), which was prevented by knockdown of rictor in hepatoma cell line HepG2 and colon cancer cell line SW480.

#### 3.3.8. Downstream of mTORC2

The AGC family of kinases, including Akt, PKCα, and SGK1, is composed of substrates of mTORC2 and facilitates mTORC2 modulation of the actin cytoskeletal structure, cell survival, and proliferation [81]. Activation of mTORC2 was found to stimulate phosphorylation of protein kinase C α (PKCα) on S657. Knockdown of rictor and mTOR in HeLa cells both resulted in thick cytoplasmic actin fibers and less cortical actin staining than the control, like the knockdown of PKCα [81,126]. In addition, cell survival and proliferation are mediated by mTORC2 via priming Akt for activation through phosphorylation on S473, leading to full activation of Akt by phosphorylation on T308 by PDK1 [127]. Studies have demonstrated that depletion of rictor, mLST8, or mSIN1 of mTORC2 resulted in the ablation of the Akt phosphorylation on S473 [128,129,130].

#### 3.3.9. mTORC2 Regulation of Glycolysis

Much like mTORC1, mTORC2 is involved in the regulation of glycolysis but through distinct mechanisms. Hagiwara et al. [131] investigated the involvement of mTORC2 in glycolysis using mice with liver-specific knockout of rictor (LiRiKO mice). Livers from the LiRiKO mice showed a significant reduction in Akt phosphorylation on S473 and T450, both mTORC2 phosphorylation sites, but loss of rictor did not affect the phosphorylation of T308, the PDK phosphorylation site. Furthermore, glycolysis appeared to be negatively affected by the loss of rictor as several genes involved in glycolysis (glucokinase, pyruvate kinase, ChREBP) were down-regulated, as well as the protein expression and enzymatic activity of glucokinase. By introducing a constitutively active form of Akt, the effects caused by the loss of rictor were reversed, indicating that mTORC2 mediated glycolysis in an Akt-dependent manner. Masui et al. [132] examined the role of mTORC2 in glycolysis in glioblastoma (GBM) through c-Myc, a critical regulator of cancer cell metabolism. Rictor shRNA knockdown in GBM cells reduced the expression of c-Myc as well as genes involved in glycolysis (*Ldha*, *Hk2*, *Pdk1*, *Eno1*, *Glut1*) and the pentose phosphate pathway (PPP) (*G6pd*, *Pgd*, *Rpe*, *Rpia).* The down-regulation of these genes was concomitant with dramatically reduced glucose consumption, lactate production, glutamine uptake, and glutamate secretion. In addition, mTORC2 was shown to regulate c-Myc expression by promoting the phosphorylation and inactivation of histone deacetylases, consequently allowing acetylation of the transcription factors FOXO1 and FOXO3 that alleviated the miR-34c-dependent repression of c-Myc. Unlike rictor knockdown, shRNA knockdown of raptor or Akt had only a modest effect on c-Myc expression. This also indicated that mTORC2 could regulate glycolysis independently of Akt. More recently, Zu et al. [133] examined the involvement of mTORC2 in a c-Myc-driven mouse model of hepatocellular carcinoma (HCC), in which ablation of rictor prevented the development of HCC in vivo. These mice also show an up-regulation of phosphorylated Akt on S473 compared to livers of wild-type mice, suggesting the c-Myc may also regulate mTORC2. Much like rictor ablation, ablation of Akt1, but not Akt2, also prevented HCC in vivo. This study indicated that mTORC2 and Akt1 are required for c-Myc-driven HCC. Together, these studies highlight that mTORC2 regulates glycolysis through activation of Akt and c-Myc, depending on the context. This contrasts with mTORC1 that regulates glycolysis via other factors including HIF1α and SREBP1.

#### 3.3.10. Role of mTORC1 and mTORC2 in Immunity

Both mTORC1 and mTORC2 are involved in innate and adaptive immunity. The mTOR pathways are essential for many immune functions that include suppression of IL-12 and IL-23 production, enhancing M2 macrophage polarization, antigen presentation, and innate immune cell migration. Activation of mTOR in immune cells and other cell types within the tumor microenvironment also affects cancer progression through supporting angiogenesis, metastasis, and drug resistance. The immunological roles of mTOR signaling and its involvement in the tumor microenvironment will not be discussed in this review but have been reviewed in detail elsewhere [134,135].

### 3.4. mTOR Pathway Inhibitors: Mechanisms of Action and Efficacy in Clinical Studies for the Treatment of HER2-Positive Breast Cancer

#### 3.4.1. Rapamycin (Sirolimus) and Rapalogs

Rapamycin (sirolimus) is the first mTOR inhibitor discovered as a naturally occurring compound purified from the bacterium *Steptomyces hygroscopicus.* Rapamycin and its analogs, or rapalogs, inhibit mTORC1 kinase activity by binding to the small mTOR-binding protein FK506-binding protein 12 (FKBP12), and then irreversibly binding to the FRB domain of mTOR, thereby inhibiting the kinase activity of the adjacent catalytic domain [136,137]. In contrast to mTORC1, the mTOR subunit of mTORC2 is insensitive to rapamycin; however, prolonged treatment can disrupt mTORC2 assembly in certain cell types whereby the mTOR protein is unavailable for assembly into mTORC2 as it is sequestered in a complex with rapamycin [138] (reviewed in [139]).

The synthetically made rapalogs were designed to improve the pharmacokinetic properties relative to rapamycin. Like rapamycin, rapalogs are metabolized by the liver, and in the intestine via cytochrome P450 enzyme CYP-3A4 [140]. The rapalogs’ metabolites are eliminated mainly through the gastrointestinal tract (reviewed in [141]). Temsirolimus (CCI-779) is one of the first rapalogs and a rapamycin prodrug developed with a higher water solubility than rapamycin to allow intravenous injection and oral administration [142]. Everolimus (EVE; RAD001) is a non-prodrug designed to have improved oral bioavailability and a shorter half-life compared to rapamycin [143]. Everolimus also inhibits mTORC1 kinase activity and activation of its downstream effectors S6K1 and 4E-BP1 as well as the expression of the proangiogenic transcription factor hypoxia-inducible factor 1a (HIF-1a) [144,145]. Ridaforolimus (AP23573, deforolimus) is a non-prodrug rapalog that is designed to combine the improvements of temsirolimus and everolimus whereby water solubility, chemical stability, and bioavailability are improved compared to rapamycin and can be administered orally or intravenously [146]. One of the more serious issues with rapamycin and rapalogs is the induction of Akt and ERK signaling in cancer cells (Figure 1) that is caused by activation of mTORC2 due to mTORC1 inhibition, as discussed previously [111,147] (for further details, see review [148]). Rapamycin treatment of breast cancer cell lines MCF-7 and MDA-MB-468 and the prostate cancer cell line DU-145 (PTEN wild type) all resulted in a significant up-regulation of Akt phosphorylation on S473 [147]. Akt activation in cancer cells is an unfavorable effect of mTORC1 inhibition as Akt promotes survival and proliferation [147]. Hence, mTORC1 inhibition stimulating Akt and ERK signaling is an unfavorable effect in cancer therapy as this promotes tumor survival and proliferation [111].

#### 3.4.2. mTOR Kinase Inhibitors (TKIs)

Second-generation mTOR kinase inhibitors (TKIs) act as competitive ATP inhibitors by binding to the kinase domain of the mTOR subunit present in both mTORC1 and mTORC2, consequently blocking their catalytic activity. Furthermore, TKIs have the advantage over rapamycin and rapalogs as TKIs block the feedback activation of PI3K-Akt signaling resulting from mTORC1 inhibition [139]. Torin-1 is a TKI highly selective for mTOR and does not affect the stability of mTOR complexes [149]. AZD8055 inhibits the mTOR subunit of both mTORC1 and mTORC2, and class I PI3K isoforms [150,151]. Furthermore, AZD8055 treatment impaired cell proliferation of several cancer cell lines and impaired tumor growth in xenograft mouse models [150]. AZD2014 (vistusertib), an analog of AZD8055 with improved properties, is also a potent ATP-competitive inhibitor selective for mTORC1 and mTORC2 [152]. AZD2014 was shown to inhibit the phosphorylation of mTORC1 and mTORC2 substrates in vitro and in vivo. Although everolimus was found to be a more potent inhibitor of mTORC1 activity than AZD2014, cell proliferation was inhibited more effectively by AZD2014. In xenograft mice implanted with MCF-7 cells, an ER-positive and PR-positive breast cancer cell line, treatment using AZD2014 or the selective estrogen receptor degrader fulvestrant and the combination of the two drugs were assessed. The combination of both AZD2014 and fulvestrant was found to be more effective in inhibiting tumor growth than either drug alone, and patients receiving this combination treatment presented with low toxicities [152]. Although TKIs are an improvement compared to rapamycin and its analogs, they have shown a minimal effect in reducing lung tumor growth in mice with mutant K-Ras [153], and there is greater toxicity associated with these drugs [154]. In the MANTA phase II clinical trial, treatment of ER+ breast cancer patients using fulvestrant plus vistusertib therapy was less effective as PFS was lower compared to fulvestrant plus everolimus therapy [155]. Prolonged inhibition of mTORC2 was found to increase Akt phosphorylation on T308 without the presence of Akt phosphorylation on S473, suggesting a possible mechanism of resistance [139]. A phase I clinical trial assessed the therapeutic use of another dual mTORC1/2 inhibitor, CC-223, for patients with advanced solid tumors or multiple myeloma [156]. CC-223 was well-tolerated with a low number of grade 3 toxicities and other manageable toxicities. In phase II clinical trials using patients with non-pancreatic neuroendocrine tumors, CC-223 therapy achieved a median PFS of 19.5 months, an ORR of 7.3%, and a disease control rate of 90.2%, and a tumor size reduction of any magnitude was observed in 73.2% of patients [157]. This demonstrated that dual mTORC1/2 inhibition may be an effective treatment for this and other types of cancer.

#### 3.4.3. Dual PI3K-mTOR Inhibitors

The dual PI3K-mTOR inhibitors are small-molecule inhibitors related to TKIs that bind to the ATP-binding sites of PI3K and mTOR with similar affinity. NVP-BEZ235 inhibits both mTOR and PI3K signaling in vitro and in vivo [151,158]. Furthermore, NVP-BEZ235 has been shown to suppress the growth of hypopharyngeal squamous cell carcinoma (HSCC) cell line FaDu in vitro and in xenograft mouse models. However, clinical trials observed that NVP-BEZ235 therapy was associated with high toxicity and little to no clinical improvement, leading to discontinuation of the treatment [159,160,161]. PI-103 is an inhibitor of mTORC1, mTORC2, DNA-PK, and several PI3K isoforms. Thus far, PI-103 was demonstrated to inhibit the proliferation of various cancer cell lines and tumor growth in xenograft mouse models [162,163].

### 3.5. Clinical Response of mTOR Inhibition in HER2-Positive Breast Cancer Patients

Clinical trials have assessed the efficacy of rapalogs in combination with trastuzumab for the treatment of HER2-positive breast cancer. In a phase I study, everolimus, trastuzumab, and paclitaxel combination therapy of HER2-positive breast cancer showed promising results with an ORR of 44% and median PFS of 34 weeks (Table 2); however, grade 3 to 4 neutropenia was common in patients [164]. In the BOLERO-1 phase III clinical trial [165,166], everolimus, trastuzumab, and paclitaxel combination therapy for HER2-positive breast cancer showed an objective response, and median PFS was similar between patients who received the addition of everolimus to the trastuzumab and paclitaxel therapy compared to the addition of the placebo. Interestingly, patients with HER2-positive/hormone receptor-negative (HR−) status showed an improved median PFS compared to the placebo group (20.27 months vs. 13.08 months) (Table 2). In the BOLERO-3 phase III clinical trial [167], everolimus, trastuzumab, and mitotic inhibitor vinorelbine combination therapy of patients with trastuzumab-resistant, HER2-positive breast cancer showed minor improvements in response and median PFS (Table 2). In both BOLERO-1 and BOLERO-3 trials, toxicity was more frequent for patients receiving everolimus than in the placebo groups. Another rapalog, ridaforolimus, used in combination with trastuzumab therapy showed some success in clinical benefit rates and median PFS (Table 2) with modest toxicities [168]. In the LCCC 1025 phase II clinical trial [169], everolimus, trastuzumab, and vinorelbine combined therapy for HER2-positive breast cancer patients with brain metastases showed similar clinical benefit rates relative to other clinical studies using everolimus for HER2-positive breast cancer therapy (Table 2). Together, these clinical trials observed that combining rapalogs and trastuzumab presented a slight improvement in clinical benefits compared to HER2 inhibition alone. However, mTORC1 inhibition resulted in a higher incidence of low-grade and grades 3–4 toxicities. Furthermore, these studies assessed the efficacy of rapalogs that inhibit mTORC1 and not mTORC2 [170,171]. Inhibition of mTORC1 promotes mTORC2 activation and hence mTORC2-dependent feedback activation of the Akt signaling pathway [127,171], thus promoting cancer cell survival and proliferation. Interestingly, in preclinical studies and in vitro, O’Brien et al. demonstrated that combined inhibition of Akt and mTOR prevented the feedback activation induced by trastuzumab-mediated inhibition of mTORC1 and sensitized trastuzumab-resistant cells to trastuzumab [172].

For the dual PI3K-mTOR inhibitors, clinical trials assessing the efficacy of NVP-BEZ235 therapy for prostate cancer [159], renal cell carcinoma [160], and solid tumors [161] observed high toxicity and no improvement in clinical responses. More recently, the maximum tolerated dose of NVP-BEZ235, as well as formulations and dosage forms, were assessed in patients with solid tumors, including those with HER2-positive breast cancer [173]. Here, Rodon et al. reported that the adverse effects of NVP-BEZ235 monotherapy, in any formulation, and when used in combination with trastuzumab, were similar. The authors noted that the onset of the adverse effects occurred shortly after dosing and may be caused by low absorption and precipitation of the drug at high doses rather than mechanism-based toxicities. Furthermore, certain toxicities associated with PI3K inhibition (hyperglycemia and rash) and mTOR inhibition (pneumonitis) were not observed [173]. Despite this, an anti-tumor effect with a favorable safety profile could not be achieved with any NVP-BEZ235 treatments or formulations.

This review focuses on the clinical efficacy and toxicity of mTOR inhibitors in HER2-positive breast cancer, but these have been assessed in many other cancers. For further details of the clinical efficacy of mTOR inhibitors in cancers other than HER2-positive breast cancer, please refer to reviews: [174,175].

**Table 2 cancers-13-02922-t002:** Clinical trials targeting mTOR in HER2-positive breast cancer.

Treatment		Trial Name	Patients	ORR	Median PFS	OS (Months)	REF
Everolimus	EV, V, T	NCT0046556 Phase Ib/II	33	44.00%	34 weeks	n/a	[164]
	EV, EX	NCT00426530 Phase Ib	50	19.10%	30.7 weeks	n/a	[176]
	EV, P, T	BOLERO-1 (NCT00876395) Phase III	719	67.00%	14.95 months	63% survival at 41.3	[165]
			303 (Asian subset)	69.20%	18.40 months, 25.46 months (HR-)	n/a	[166]
	EV, P, T	BOLERO-3 (NCT01007942) Phase III	569	41.00%	7 months	n/a	[167]
	EV, P, T	LCCC 1025 (NCT01305941) Phase II	32	CBR: 20% (>6 months), IRR: 4%	3.93 months (TTP)	12.2	[169]
Ridaforolimus	R, T	NCT00736970 Phase IIb	34	CBR: 34.3%	5.4 months	17.7	[168]
Dactolisib (BEZ235)	B	NCT00620594 PhaseI/Ib	153	0%	SD: 31.1–42.4%	n/a	[173]
	B, T		30	13.30%	SD: 40%	n/a	

CBR = clinical benefit rate, TTP = time to intercranial progression, IRR = intracranial response rate, SD = stable disease, EV = everolimus, EX = exemestane, P = pacitataxel, T = trastuzumab, V = vinorelbine, B = BEZ235, HR = hormone receptor.

### 3.6. Preclinical Studies of mTOR Pathway in HER2-Positive Breast Cancer Mouse Models

Genetically engineered mouse models of HER2-positive breast cancer overexpress Neu/HER2 (wild-type or mutant Neu) in mammary glands under the Mouse Mammary Tumor Virus (MMTV) promoter. The first Neu/HER mouse model, MMTV-*neu* mice, was initially described in 1988 by the Leder laboratory [177]. MMTV-*neu* mice constitutively express Neu/HER2 and produce rapidly growing, highly metastatic mammary tumors [177] (reviewed in [178]). Later, in 2008, MMTV-NIC (*neu-IRES-Cre*) mice were generated, which simultaneously expresses *neu* and *cre* recombinase (activated Neu/HER2-MMTV-Cre) under the endogenous *Erbb2* promoter. NIC mice produce aggressive HER2-positive mammary tumors at around 146 days old [179].

When human breast cancer tissue microarrays were analyzed for expression of LKB1, a critical negative regulator of mTORC1, 31% of HER2-positive breast cancers were deficient in LKB1 expression [77]. Based on those observations, *Lkb1* (also known as *Stk11*) was knocked out in NIC mice [179], by genetic crossing of NIC mice with *Stk11^fl/fl^* mice, to generate *Stk11^−/−^*NIC (*Lkb1^−/−^*NIC) mice. The loss of *Lkb1* reduced the latency of tumorigenesis in *Lkb1^−/−^*NIC mice compared with NIC mice [77]. Furthermore, tumors from *Lkb1^−/−^*NIC mice showed enhanced phosphorylation of the S6K1 substrate ribosomal protein (S6), elevated ATP levels, and changes in metabolic enzymes and metabolites indicative of mTORC1 hyperactivation (Figure 2) [77]. Chen et al. [180] assessed the role of LKB1 in breast cancer, including HER2-positive breast cancers, using immunohistochemical analysis of tumors from early breast cancer patients and in silico analysis obtained from the Molecular Taxonomy of Breast Cancer International Consortium (METABRIC) dataset. Although LKB1 protein expression did not correlate with HER2 status, high LKB1 expression in HER2-positive breast cancer patients correlated with improved overall survival, consistent with the findings of Andrade-Vieira et al. [77] that found NIC mice expressing LKB1 had an increased tumor latency compared to *Lkb1^−/−^*NIC mice.

Using the mammary epithelial cells (MECs) of tumors isolated from both from *Lkb1^−/−^*NIC and NIC mice, treatments using mTOR inhibitors (rapamycin, Torin-1, and AZD8055) reduced phosphorylation of S6; however, Torin-1 and AZD8055 reduced the levels of phosphorylated Akt on both S473 and T308 (Figure 2) [77], indicating both mTORC1 and mTORC2 were inhibited. Both NVP-BEZ235 and AZD8055 treatments of *Lkb1^−/−^*NIC mice reduced the tumor volume compared with the vehicle control, though AZD8055 treatment inhibited the tumor volume significantly more compared with NVP-BEZ235 treatment [151]. When considering mitochondrial function, AZD8055 treatment reduced the mitochondrial content of primary mammary tumors isolated from *Lkb1^−/−^*NIC mice, but not in mammary epithelial cells isolated from control wild-type female littermates. Tumors isolated from AZD8055-treated *Lkb1^−/−^*NIC mice showed significantly reduced expression of glycolytic enzymes (hexokinase 2, lactate dehydrogenase (LDH), and pyruvate dehydrogenase (PDH)) and phosphorylation of S6 compared with the vehicle control. However, tumors from AZD8055-treated mice also showed strong induction of ERK and p90RSK phosphorylation (Figure 2). This activation of the MAPK/ERK pathway is indicative of a pro-survival feedback loop often observed with mTOR inhibition that can contribute to cancer recurrence [151] (reviewed in [181]).

## 4. Glycolysis in Breast Cancer

### 4.1. Glycolysis and Oxidative Phosphorylation

Glycolysis uses glucose to generate two molecules of pyruvate and energy in the form of ATP (Figure 3) [182] (reviewed in [183]). Under aerobic conditions, pyruvate is transported into the mitochondria and converted to citrate and CO_2_. Citrate goes on to the tricarboxylic acid (TCA) cycle that facilitates the transport of electrons to the electron transport chain (ETC). Electrons generated by the ETC are used for oxidative phosphorylation (OXPHOS) to generate ~36 ATP molecules per glucose molecule (reviewed in [184]). When oxygen is limited, pyruvate is metabolized via anaerobic glycolysis, which generates ATP less efficiently but 100X more rapidly compared to OXPHOS [185]. In anaerobic glycolysis, LDH catalyzes the reduction of pyruvate and regeneration of NAD^+^, where pyruvate and NADH are converted to lactate, NAD^+^, and two ATP molecules (Figure 3).

### 4.2. Glycolysis in Cancer

Warburg et al. [186] made the landmark observation that tumor cells metabolized high levels of glucose to produce ATP and lactate in the presence of oxygen. The metabolic shift of OXPHOS to aerobic glycolysis in tumor cells is known as the “Warburg effect”. This established the paradigm that cancer cells use glycolysis instead of OXPHOS to produce ATP due to impaired OXPHOS function, potentially due to mitochondrial damage. However, contrary to the hypotheses postulated since the discovery of the Warburg effect, mitochondria in most tumor cells are intact and OXPHOS is not impaired in most cases [187]. Recent evidence indicates that OXPHOS and aerobic glycolysis do function in cancer cells [188]. Using a population of cancer stem cells (CSC) from epithelial ovarian cancer, Pasto et al. [189] found that the expression of many essential enzymes involved in TCA and ETC were up-regulated (e.g., citrate synthase (CS), isocitrate dehydrogenase (IDH2), ATP5B, HKII, PKM, and PFK) concomitant with the up-regulation of GLUT1 cell surface levels in CSCs (CD44^+^CD117^+^) compared to non-tumorigenic CD44^+^CD117^−^ cells. The CSCs showed characteristics consistent with cells that use OXPHOS such as elevated generation of mitochondrial reactive oxygen species (ROS), increased membrane potential, and ETC inhibition, resulting in apoptosis. Through evaluation of metabolites isolated from three types of *Kras^G12D^*-driven non-small cell lung cancer (NSCLC) mouse models and xenograft models using A549 and H1975 lung cancer cell lines, Davidson et al. [188] observed that the metabolism of lung tumors relied on oxidative metabolism and aerobic glycolysis. Tumors from NSCLC mice showed increased glucose uptake concomitant with increased TCA cycle metabolism. Increases in the levels of lactate were observed in lung tumors of two of the three NSCLC models and the tumors of xenograft mice relative to normal tissues. Furthermore, tumor growth rates differed between the lung tumors from the two NSCLC models, although they both showed high levels of lactate, which argues that lactate levels do not necessarily correlate with proliferation. Interestingly, lung tumor cells isolated from one of the NSCLC mouse models relied more on glutamine metabolism to support cell growth, whereas tumor cells analyzed directly from the same NSCLC mouse model did not rely on glutamine metabolism. The observation of metabolic differences between lung cancer cells in vitro and in vivo suggested a role of the tumor microenvironment in influencing cancer cell metabolism. Interestingly, OXPHOS supports tumor growth through the metabolites generated by glycolysis and the ETC. For example, the generation of aspartate, a precursor used to synthesize purine and pyrimidine, can be metabolized by tumor cells to support growth [190,191]. Moreover, several studies found that the loss of OXPHOS by depletion of mitochondrial DNA (mtDNA) resulted in reduced tumor cell proliferation in vitro and tumorigenicity in vivo [192,193,194].

Several studies have observed that resistance to targeted therapies in cancer cells is associated with increased glycolytic activity and expression of glycolytic enzymes [195,196,197]. The concept of metabolic reprogramming has gained popularity as a means for tumors to adapt to the metabolic requirements for survival (for further details, please refer to reviews [198,199]).

### 4.3. Glycolysis in HER2-Positive Breast Cancer and the Involvement of mTOR

HER2 overexpression in breast cancer cell lines increased glycolysis as indicated by increased glucose uptake and lactate production, and decreased oxygen consumption rates [200]. Zhao et al. [200] found that HER2 overexpression in breast cancer cell lines MCF-7 and MDA-MB-435 increased glycolysis, indicated by increased glucose uptake and lactate production, and decreased oxygen consumption rates. Of the HER2-positive breast cancer subtypes, metabolomics data of patient tumors showed that the HER2-E and basal-like subtypes had elevated levels of glycolytic enzymes G-6-P and F-6-P and lactate compared to the luminal A and luminal B subtypes [201]. Zhang et al. [202] recently identified that the DNA repair protein Transcriptional repressor zinc finger and BTB domain containing 1 (ZBTB1) was dramatically down-regulated in breast tumors of HER2-positive patients, and that ZBTB1 suppressed HER2 expression by binding to the estrogen receptor alpha (ERα) binding site of the HER2 intron. Re-expression of ZBTB1 in HER2-expressing breast cancer cell lines reduced lactate production, glucose uptake, and down-regulated LDH and HK expression, indicating that the elevated expression of HER2 due to the loss of ZBTB1 promotes aerobic glycolysis [195]. Interestingly, Gale et al. [203] observed a metabolic shift in HER2-positive cell lines that acquired trastuzumab resistance. Here, trastuzumab-resistant, HER2-positive cells showed an up-regulation of numerous genes encoding enzymes involved in OXPHOS. Although the oxygen consumption rate was similar among parental HER2-positive cell lines and trastuzumab-resistance cells, lactate production was not examined. Trastuzumab-resistant cells and patient data showed increased expression of the ATP synthase subunits ATP5J and ATP5B, which were also found to correlate with poor survival. The addition of an ATP synthase inhibitor re-sensitized tumor cells to trastuzumab in xenograft mouse models of HER2-positive breast cancer. In contrast, studies have observed increased glycolysis, via increased glucose uptake and lactate production, in trastuzumab-resistant, HER2-positive breast cancer cells in vitro and in vivo using xenograft mouse models [195,204]. Aberrant activation of Akt, which is activation by HER2, has been observed to increase glucose consumption and lactate production, indicative of aerobic glycolysis in cancer cells [205]. Using murine-derived leukemic and human glioblastoma cell lines and xenograft mouse models, Elstrom et al. [170] demonstrated that overexpression and activation of Akt stimulated aerobic glycolysis as indicated by the increase in glucose consumption and lactate production. This effect was negated by treatment with the Akt inhibitor LY294002 in vitro [205,206] and in vivo [205]. The glycolytic activities in HER2-positive breast cancer cell lines are reduced in response to trastuzumab treatment. Together, these studies indicate that activation of the HER2-Akt pathway enhances glycolytic activity. Constitutive activation of mTORC1 was also shown to promote glycolysis. Loss of NPRL2, a negative regulator of mTORC1, decreased the levels of TCA metabolites and was concomitant with increases in glucose uptake and lactate production [207]. This suggests that mTOR hyperactivation is a critical contributor to the elevated glycolytic activity observed in HER2-positive breast cancer.

As the HER2-E subtype expresses higher levels of HER2 and glycolytic metabolites compared with other HER2-positive breast cancer subtypes, the HER2-E subtype also displayed higher levels of phosphorylated S6K, a substrate of mTORC1 [208]. Likewise, since the loss of *Lkb1* causes mTORC1 hyperactivation, tumors from the *Lkb1^−/−^*NIC mouse model of HER2-positive mammary cancer showed elevated levels of glycolytic metabolites, including lactate, and up-regulated LDH and PDH expression compared with control wild-type (WT) mice [77]. Inhibition of mTORC1/mTORC2 (Torin-1, AZD8055) or mTORC1 (rapamycin) down-regulated LDH expression and had little to no effect on PDH expression in the primary tumor cells isolated from *Lkb1^−/−^*NIC mice compared to the cells from NIC mice [77,151]. Furthermore, tumors from AZD8055-treated *Lkb1^−/−^*NIC mice showed reduced glycolytic activity and oxygen consumption rates compared with vehicle-treated mice. Characterization of mitochondrial content, size, and cristae density was greater in mammary tumors from *Lkb1^−/−^*NIC mice compared with mammary glands from control WT mice. AZD8055 treatment reduced the mitochondrial content in *Lkb1^−/−^*NIC but not in control WT mice, indicating the role of mTOR activity in mitochondrial biogenesis in HER2-positive cancer [151]. This study demonstrates that therapies that simultaneously target mTORC1/mTORC2 and glycolytic metabolism in cancer produce the best therapeutic outcome against HER2-positive breast cancer.

### 4.4. Glucose Analogs and the Effect on Cancer Cells

Tumor cells can develop a dependency on glycolysis for survival. Glucose analogs cause glucose deprivation, resulting in the suppression of glycolysis as they cannot be metabolized by cells. 2-deoxy-D-glucose (2-DG) is a glucose analog that is taken into the cytosol through glucose transporters (GLUTs), where hexokinase phosphorylates 2-DG to generate 2-DG-P; however, phosphohexose isomerase is not able to metabolize 2-DG-P any further (reviewed in [209]). Here, downstream glycolysis and production of cellular ATP are inhibited by the accumulation of 2-DG, which is associated with impaired cell cycle progression and enhanced cell death of tumor cells [210]. Although 2-DG negatively affects cell cycle progression, studies have demonstrated that the inhibition of glycolysis by 2-DG monotherapy is concomitant with the induction of Akt phosphorylation at T308 and S473 [211,212]. This could have negative implications in the efficacy of 2-DG as a cancer treatment as 2-DG-induced Akt activation would oppose the 2-DG-dependent inhibition of cell proliferation and survival.

Considering that the availability of endogenous glucose can mitigate the efficacy of 2-DG, 2-DG may not completely block glycolysis. However, antiproliferative and cell death-promoting effects of 2-DG have been observed in vitro and in vivo in cancer cells [213]. Treatment using 2-DG induces endoplasmic reticulum stress, leading to autophagy. This results from the accumulation of misfolded proteins in the ER lumen concomitant with ER stress and the unfolded protein response, a mechanism of inhibiting protein translation to relieve ER stress [213]. Inhibition of autophagy prevented 2-DG-induced autophagy and ER stress but did not reverse the depletion of ATP. Furthermore, treatment using oxamate, which depletes ATP without inducing ER stress, did not induce autophagy [213], indicating that 2-DG induces autophagy in cancer cells by increasing ER stress and not by ATP depletion.

Suppression of glycolysis in HER2-positive breast cancer has been observed to reduce HER2-driven mammary tumor cell growth in vitro as well as in vivo with mouse models [214]. In the *Lkb1**^−/−^*NIC mouse model of HER2-positive mammary cancer, 2-DG monotherapy reduced tumor burden and growth compared with vehicle-treated mice, but to a lesser extent than AZD8055 monotherapy [151]. Tumors from 2-DG-treated mice showed reduced glycolysis, oxygen consumption rate, mitochondrial content, and down-regulation of HK, PDH, and LDH expression. The combination of AZD8055 and 2-DG further augmented these effects, indicating that 2-DG sensitized the tumor cells to mTORC1/2 inhibition. In addition, 2-DG treatment did not significantly affect the phosphorylation status of S6, ACC, AMPK, p90RSK, and ERK compared to vehicle-treated mice. Interestingly, 2-DG induced phosphorylation of AMPK (T172) concomitant with reduced mTORC1 activity as observed from reduced phosphorylation of mTOR (S2448) and S6K1 (T389) in various cell types (Figure 1) [215]. Hurley et al. reported that 2-DG induced the phosphorylation of AMPK and ACC in HeLa cells that were LKB1^+/+^ or LKB1^−/−^ [216], indicating that 2-DG induced another kinase responsible for AMPK phosphorylation and activation independent of LKB1. Inhibition of CaMKK2, which also phosphorylates AMPK on T172, using STO-609 prevented 2-DG-induced AMPK phosphorylation [216,217]. Furthermore, since HeLa cells do not express the CaMKKβ isoforms 1 and 2 [218], shRNA knockdown of the CaMKKβ3 isoform, but not the CaMKKα isoform, prevented AMPK phosphorylation by 2-DG. Since 2-DG did not significantly induce AMPK or ACC phosphorylation in tumors from *Lkb1**^−/−^*NIC mice [151], this may suggest CaMKKβ3 is not active in this model. Together, these studies suggest that, in addition to inhibiting glycolysis directly through inhibition of phosphohexose isomerase activity, 2-DG inhibits mTORC1-mediated promotion of glycolysis.

### 4.5. Adverse Effects of 2-DG

Studies assessing the efficacy of 2-DG and other metabolic interventions in combination with various anticancer therapies have observed tolerable adverse effects from the addition of 2-DG [219] (reviewed in [220]). Preclinical studies determined that high dosages negatively impacted the respiratory frequency and mean arterial blood pressure [221], and reversible cardiac toxicity has been associated with 2-DG treatment in rats [222]. Despite the therapeutic potential of 2-DG for cancer treatment, studies have shown little to no effect of 2-DG treatment alone in inhibiting tumor growth in preclinical mice models, including genetic [151] and xenograft models [223]. In a clinical trial conducted in patients with a variety of solid tumors, Raez et al. [219] observed that patients treated with 2-DG presented with reversible hyperglycemia (100%), gastrointestinal bleeding (6%), and cardiac abnormalities (QTc prolongation; 22%).

### 4.6. Combination Therapy of mTOR Inhibition and 2-DG in HER2-Positive Breast Cancer Mouse Models

As previously discussed, HER2 hyperactivity promotes mTOR pathway-dependent up-regulation of glycolytic enzymes, indicating that mTOR is a major contributor to the enhanced glycolytic activation observed in HER2-positive breast cancer. Inhibition of the mTOR pathway can negate the effect of HER2 hyperactivity, enhancing glycolysis in cancer cells, but prolonged inhibition of mTOR pathway activity leads to an up-regulation of ERK activation [111,147]. Consistent with this, our laboratory observed an up-regulation of ERK phosphorylation, concomitant with a decrease in glycolytic activity and tumor growth, in mammary tumors from *Lkb1**^−/−^*NIC mice treated with the mTORC1/2 inhibitor AZD8055 [151]. Furthermore, mammary tumors from AZD8055-treated mice showed up-regulation of AMP, ACC, and p90RSK phosphorylation. Like mTOR inhibition, 2-DG also suppresses glycolysis but activates PI3K [211,212] and AMPK [216,217]. In our preclinical study, 2-DG reduced tumor growth to a lesser extent than AZD8055. As AZD8055 did not completely block glycolytic activity in mammary tumors of *Lkb1**^−/−^*NIC mice, combination treatment was performed by adding 2-DG to the AZD8055 treatment to further inhibit glycolysis in addition to mTORC1/2. This combination treatment further reduced the mammary tumor volume and burden compared to AZD8055 or 2-DG monotherapies, but more importantly, the addition of 2-DG blocked the AZD8055-dependent induction of ERK phosphorylation, as well as the phosphorylation of AMPK, ACC, and p90RSK [151]. Sun et al. [224] also observed that 2-DG treatment of lung cancer cells resulted in a time- and dose-dependent down-regulation of ERK phosphorylation in an LKB1/AMPK-dependent manner. Although our mouse model lacks LKB1 expression, 2-DG can induce AMPK independently of LKB1, as previously discussed in Section 4.4 of this review. These studies demonstrate the benefit of combining glycolytic inhibition via 2-DG with mTORC1/2 inhibition, in which the addition of 2-DG prevents the mTOR inhibitor-dependent induction of pro-survival MAPK pathway signaling. Furthermore, Cheng et al. showed 2-DG sensitized 5-fluorouracil (5-Fu)-resistant pancreatic cancer cell lines to the cytotoxic effects of 5-Fu [225]. They revealed that 2-DG-mediated activation of PI3K-Akt was required to sensitize the drug-resistant cells to cell growth inhibition and apoptosis induced by 5-Fu. In the study by Ma et al. [226], 2-DG was also found to overcome chemo-resistance due, in part, to the suppression of ATP-dependent drug efflux transporters. These studies strongly suggest that the addition of 2-DG may improve the efficacy of therapies using mTOR inhibition, as well as other targeted therapies, by migrating factors that contribute to drug resistance such as activation of pro-survival signaling and drug efflux.

## 5. Concluding Remarks

The progression and recurrence of HER2-positive breast cancer are supported by HER2-Akt-mTOR signaling that promotes pro-survival signaling and cell cycle progression. Additionally, the mTOR pathway in HER2-positive breast cancer mediates a metabolic shift from glycolysis to OXPHOS to further facilitate the nutrients required for tumor growth. Monotherapeutic inhibition of these signaling pathways has had modest clinical benefits, but combination therapies using mTOR and/or glycolytic inhibition show encouraging outcomes. Other combination therapies showing promising results include the use of cyclin-dependent kinase 4/6 (CDK4/6) inhibitors that block cell progression and mTORC1 activity [227]. This is particularly relevant in the HER2-E subtype that displays DNA amplification of *cyclin D1* and *CDK4* [208]. Even still, the combination of CDK4/6 and HER2 inhibitors is met with the development of drug resistance in HER2-positive breast cancer treatment [228]. Consistent with this, findings from Finn et al. also indicated that the non-luminal subtypes of HER2-positive breast cancer are more resistant to the beneficial effects of the CDK4/6 inhibitor palbociclib [229]; however, this drug has been associated with various dermatological toxicities such as alopecia (reviewed in [230]). Understanding the relationship between the HER2-Akt-mTOR pathway, glycolysis, and CDK4/6 may pave the way for improved therapies to treat HER2-positive breast cancer and overcome drug resistance.

## Figures and Tables

**Figure 1 cancers-13-02922-f001:**
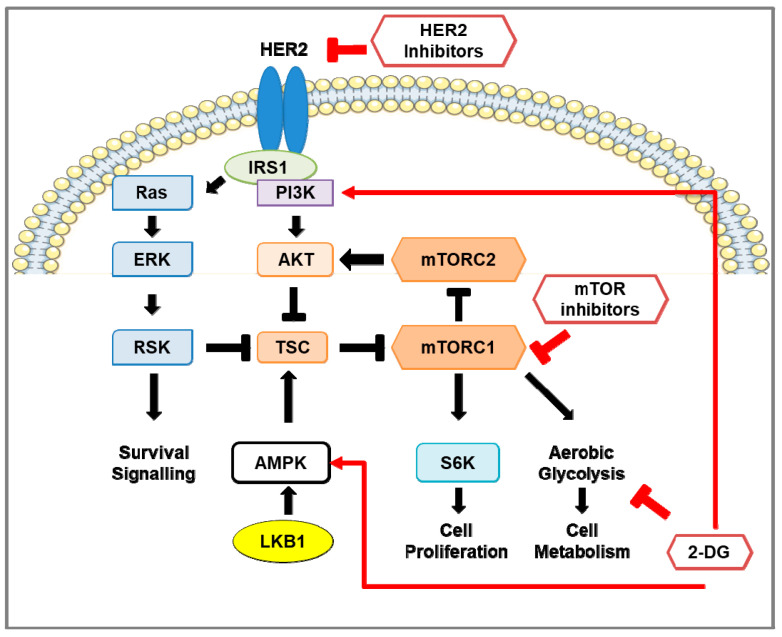
Pro-survival signaling in HER2-positive breast cancer and the effect of inhibition of HER2, mTOR, and glycolysis. Schematic representation of signaling pathways in HER2-positive breast cancer cells. HER2 and insulin receptor substrate (IRS) stimulates the RAS-ERK-RSK and Akt-mTOR pathways. Activation of mTOR signaling phosphorylates S6 kinase 1 (S6K) and eukaryotic initiation factor 4E-binding protein 1 (4E-BP1; not shown). The LKB1-AMP-dependent protein kinase alpha (AMPK) pathway negatively regulates mTOR signaling by activation of the mTOR inhibitors tuberous sclerosis complex-1 and -2 (TSC1/2). In HER2-positive breast cancer, HER2 inhibitors reduce the activity of the RAS-ERK-RSK and Akt-mTOR pathways and down-regulate glycolytic activity. However, re-activation of the mTOR pathway occurs, consequently enhancing glycolysis, pro-survival, and cell proliferation pathways. Treatment using mTOR inhibitors, such as rapamycin and rapalogs, in HER2-positive breast cancer prevents phosphorylation of S6K, inhibits cell proliferation, and reduces aerobic glycolysis through down-regulation of glycolytic enzymes. Prolonged inhibition of mTOR activates pro-survival pathways in a pERK-RSK- and Akt-dependent manner. The glucose analog 2-deoxy-D-glucose (2-DG) suppresses aerobic glycolysis, activates PI3K signaling, and promotes phosphorylation of AMPK on T172 to inhibit mTOR signaling.

**Figure 2 cancers-13-02922-f002:**
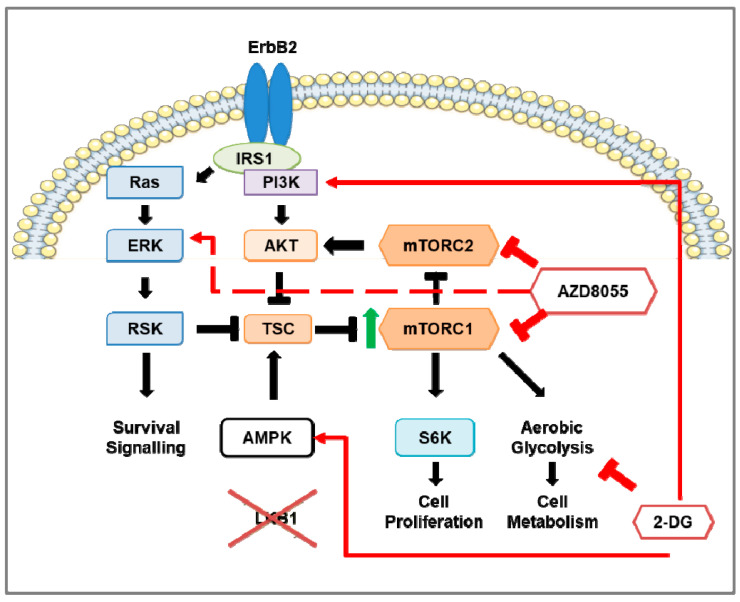
The effects of mTOR and inhibition of mTOR and aerobic glycolysis in *Stk11^−/−^*NIC (*Lkb1^−/−^*NIC) mouse model of HER2-positive breast cancer. The overexpression of ErbB2 in the breast tumors of the HER2-positive breast cancer mouse model (NIC) shows up-regulation of ERK, Akt, and mTOR signaling as well as aerobic glycolysis; however, deficiency of LKB1 expression in NIC mice (*Lkb1**^−/−^*NIC) leads to hyperactivation of mTOR (indicated by the green arrow) that enhances aerobic glycolysis. In *Lkb1**^−/−^*NIC mammary gland tumors, inhibition of mTORC1 and mTORC2 via AZD8055 prevents mTORC1-dependent phosphorylation of S6K and mTORC2-dependent Akt phosphorylation on S473, inhibits cell proliferation, and reduces aerobic glycolysis through down-regulation of glycolytic enzymes. Prolonged inhibition of mTOR activates pro-survival pathways in a pERK-p90RSK-dependent manner (indicated by the red, dashed arrow). In addition, inhibition of aerobic glycolysis via 2-deoxy-D-glucose (2-DG) does not affect mTOR activity but promotes activation of AMPK and PI3K. The addition of 2-DG to AZD8055 treatment prevented the AZD8055 induction of ERK activation.

**Figure 3 cancers-13-02922-f003:**
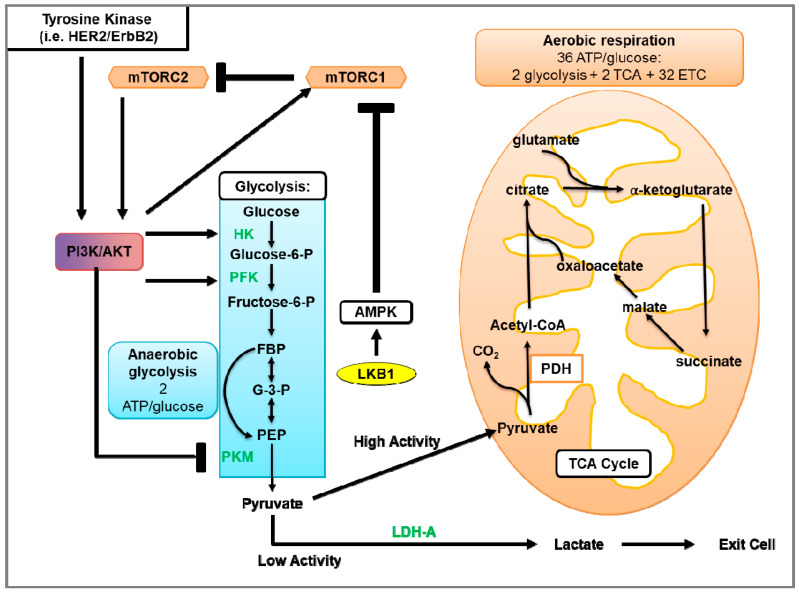
Glycolysis and aerobic respiration. The first phase of glycolysis (blue box) is the destabilization of glucose, resulting in the formation of glucose 6-phosphate (Glucose-6-P), which is further processed to β-d-Fructose 1,6-bisphosphate (FBP). In the second phase of glycolysis, FBP is converted to glyceraldehyde 3-phosphate (G-3-P). In the final phase of glycolysis, G-3-P undergoes several steps to generate pyruvate and 2 ATP molecules. In the presence of sufficient oxygen, pyruvate is transported into the mitochondria where pyruvate is processed by the tricarboxylic acid (TCA) cycle that transfers electrons to the electron transport chain (ETC) for oxidative phosphorylation, which produces ~36 molecules of ATP. In contrast, anaerobic or aerobic glycolysis instead uses LDH to facilitate the conversion of pyruvate and NADH to lactate, NAD^+^, and two ATP molecules. Growth factors mediate glycolytic metabolism through activation of tyrosine kinases (such as HER2/ErbB2) and/or Akt activation. Akt activates glycolytic enzymes such as hexokinase (HK) and phosphofructokinase (PFK). In addition to activating Akt, tyrosine kinases increase the levels of peptides phosphorylated on tyrosine residues. As pyruvate kinases M2 (PKM) binds to phosphotyrosine on peptides, this subsequently results in increased binding of phosphotyrosine peptides to PKM. This binding suppresses the PKM-mediated generation of pyruvate to be used in glycolysis. Furthermore, Akt activates the mTOR pathway, thereby stimulating protein synthesis and up-regulating numerous components of the glycolytic pathway. With low levels of ATP, LKB1-activated AMPK can inhibit mTOR-dependent protein synthesis and the activation of glycolytic genes.

**Table 1 cancers-13-02922-t001:** Clinical trials targeting HER2.

Treatment		Trial Name	Patients	ORR	Median PFS (Months)	OS (Months)	REF
Trastuzumab (T)	Monotherapy	Phase II	48	11.60%	5.1	n/a	[32]
	T, chemotherapy	n/a	90	49.50%	10	16 (from first TZB treatment)	[34]
	T, cisplatin, or carboplatin	UCLA-ORN/BCIRG Phase III	62	58%	12.7	OS not reached	[33]
Lapatinib (L)	Monotherapy	EGF20009 Phase II	136	24–31%	43% of patients at 6 months	OS not reached	[41]
	capecitabine (cap)	EGF100151 (NCT00078572) Phase III	399	13.90%	4.4	15.3	[42,43,44,45]
	L, cap			23.7%	8.4	15.6	
Neratinib (N)	Monotherapy (prior T therapy)	NCT00300781 Phase II	66	24%	22.3	n/a	[46]
	Monotherapy (no prior therapy)		70	56%	39.6	n/a	
	Monotherapy	NCT00777101 Phase II	117	29%	4.5	19.7	[47]
	N, cap	NALA (NCT01808573) Phase III	621	32.80%	5.6	24	[48]
Pertuzumab (PRT)	PRT, T	NCT03001899 Phase II	66	34%	5.5	n/a	[49]
	PRT, T, docetaxel	CLEOPATRA (NCT00567190) Phase III	808	80.20%	18.7	56.5	[50,51]
	PZB, T, chemotherapy	APHINITY (NCT01358877) Phase III	4805	n/a	94.1% of patients with IDFS at 3 years	n/a	[52]
T-DM1	Monotherapy	Phase II	112	25.90%	4.6	n/a	[53]
	Monotherapy	EMILA (NCT00829166) Phase III	991	43.60%	9.5	30.9	[54]
T-deruxtecan (DS-8201; DS)	Monotherapy (prior therapies)	DESTINY-Breast01 (NCT03248492) Phase II	184	60.90%	16.4	OS not reached	[55]
Tucatinib (TN)	TN, T, cap	HER2CLIMB (NCT02614794) Phase II	612	40.60%	7.8	44.9% at 2 years	[56]
Pyrotinib (P)	P, cap (prior therapies)	NCT01937689 Phase I	38	50.00%	35.4	n/a	[57]
	P, cap	NCT02422199 Phase II	128	78.50%	18.1	n/a	[58]
HLX02	HLX02, docetaxel	NCT03084237 Phase III	649	71.30%	11.7	n/a	[59]

IDFS = invasive disease-free survival, TTP = time to progression.

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
