# Peer review of "Targeting mTOR and Glycolysis in HER2-Positive Breast Cancer"

_cancers, 2021, doi:10.3390/cancers13122922_

Round 1

Reviewer 1 Report

The mTOR kinase activation is well-known in many cancers, there are several publications about increased mTOR hyperactivity especially in advanced breast cancers. It is also described in many reviews that mTOR could be considered as a master regulator of cellular metabolism in the signalling network. mTOR hyperactivity itself promotes the glycolysis via increasing the expression of several metabolic enzymes in these processes. Therefore, to combine mTORIs with other specific anti-metabolic treatments could have some interests. Unfortunately, mTOR inhibitors and glycolysis inhibitor therapies have several toxic side effects in anticancer treatments, especially in combinations. Moreover, combining metabolic inhibitors with different target mechanisms in cancer treatments could switch off cellular survival, disturbing cellular bioenergetics and cause metabolic catastrophe in tumour mass. Based on this, the aim of the submitted review could be interesting and summarising this subject related results based on the literature and on already closed trials, could expect high interest.

The selected topic is very interesting but the review not as precise. There are unclear imprecise parts in the abstract e.g. first and last sentences (we could not inhibit glucose). This attitude is characteristic for the whole manuscript. The authors collected the old and some but not all important novel data about three different aspects: - anti-HER2 therapy; - mTOR hyperactivity as a target; - glucose metabolism and glycolysis as potential target. The main problem is that these three parts were not analysed in their complexity and somewhere the newest clinical results e.g. the withdraw of certain dual mTOR inhibitors (e.g. BEZ) and glycolysis inhibitors based on their high toxicity are missing from the manuscript.  The main conclusion, that targeting mTOR kinase and glucose metabolism were not supported by enough data, the authors highlighted only some in vitro results and in these there are some miss-interpreted in vitro data (e.g. BT474 – HER2+ and HR+), as well. In the figures the arrows and their meanings are confusing. The cited clinical trials related to mTORIs and Glycolysis were not refreshed. The mTORC1 and C2 complexes and their different functions and inhibitory sensitivity were not mentioned in the review, however, these complexes have different metabolic functions. It would be very interesting to summarise their functions in glucose metabolism for example. The side effects of glycolysis inhibitors completely missing.  The authors need to refresh their reviewed articles in a next version, especially about mTOR and glycolysis inhibitors and their clinical phase trial reports.

Based on these, I cannot suggest to accept this review in the present form because sufficiently revise this manuscript needs more effort and time. In addition, the authors can submit this review in a new, substantially revised version because the subject of the manuscript is very interesting.

Author Response

Comment #1: The selected topic is very interesting but the review not as precise. There are unclear imprecise parts in the abstract e.g. first and last sentences (we could not inhibit glucose). This attitude is characteristic for the whole manuscript.

Revision: We have made revisions to the abstract and the rest of the manuscript to address parts that were unclear.

Comment #2: newest clinical results (e.g. the withdraw of certain dual mTOR inhibitors (e.g. BEZ) and glycolysis inhibitors based on their high toxicity) are missing from the manuscript. The authors collected the old and some but not all important novel data about three different aspects: - anti-HER2 therapy; - mTOR hyperactivity as a target; - glucose metabolism and glycolysis as potential target.

Revision: We have updated the clinical trials of mTORIs for HER2-positive breast cancer (HER2+ BC). Reviewing clinical trial websites, we were not able to find clinical trials for HER2+ BC where glycolysis was being targeted for inhibition. We expanded this section to include clinical trials where 2-DG or mTORIs are being tested for the treatment of other cancers. The following sections have been added.

  • 3.4.2. mTOR kinase inhibitors (TKIs); additional references #155-157
  • 3.4.3. Dual PI3K-mTOR inhibitors; additional references #159-161
  • 3.5. Clinical response of mTOR inhibition in HER2-positive breast cancer patients; additional references #166, #173-175
  • 4.5 Adverse effects of 2-DG; additional references #218

Comment #3: The main problem is that these three parts were not analysed in their complexity and somewhere the The main conclusion, that targeting mTOR kinase and glucose metabolism were not supported by enough data, the authors highlighted only some in vitro results and in these there are some miss-interpreted in vitro data (e.g. BT474 – HER2+ and HR+), as well.

Revision: We revised the descriptions of these studies to better clarify our discussion of these topics.

Comment #4: In the figures the arrows and their meanings are confusing.

Revision: We understand that the figures were too dense, thus we have simplified the figures and now believe they are easier to understand.

Comment #5: The cited clinical trials related to mTORIs and Glycolysis were not refreshed.

Revision: Please see Revision for Comment #2

Comment #6: The mTORC1 and C2 complexes and their different functions and inhibitory sensitivity were not mentioned in the review.

Revision: These were discussed in the original manuscript that was submitted (mTORC1 complex [page 5, lines 216-224] and its functions [Section 3.3 starting on page 7] and mTORC2 complex [page 5, lines 224-229] and functions [several mentions within Section 3.3 starting on page 7] mTORC1 and C2 sensitivities to inhibitors [Page 8, lines 338-345].

Certainly, a more detailed discussion of mTORC1/mTORC2 function was necessary as per the reviewer’s comment. This and several other additional details have been added in the latest version of the manuscript. See update #7.

Comment #7: It would be very interesting to summarise their [mTORC1 and C2] functions in glucose metabolism for example.

Revision: Although the function of mTORC1 in glycolysis was described in the original manuscript, we appreciate that overall, details were lacking. Hence, additional sections describing the roles of mTORC1 and C2 have been added to the revised manuscript. The following sections have been added:

  • 3.3.1 mTORC1 activation of S6K1 and 4E-BP1
  • 3.3.2 mTORC1 regulation of cell cycle
  • 3.3.3. mTORC1 regulation of AKT
  • 3.3.4. mTORC1 regulation of glycolysis
  • 3.3.5. mTORC1 regulation of mitochondrial biogenesis
  • 3.3.6. mTORC1 regulation of autophagy
  • 3.3.7. mTORC1 regulation of mTORC2
  • 3.3.8. Downstream of mTORC2
  • 3.3.9. mTORC2 regulation of glycolysis
  • 3.3.10 Role of mTORC1 and mTORC2 in immunity

Comment #8: The side effects of glycolysis inhibitors completely missing.

Revision: In the original manuscript, the side effects of glycolysis inhibition were mentioned, for example, 2-DG “causing reversible cardiotoxicity and hypoglycemia” (in preclinical trials) on page 17, lines 875-877. We appreciate Reviewer #1’s comments and have now included two separate sections that discuss 2-DG.

  • 4.5 Adverse effects of 2-DG
  • 4.6 Combination therapy of mTOR inhibition and 2-DG in HER2-positive breast cancer mouse models

Reviewer 2 Report

an interesting review about the role of mTOR and Glucose Metabolism in HER2-Positive Breast Cancer. I have some queries:

Page 16 line 689 after palbocinib description you should add: "This drug has been associated with various dermatological toxicities such as alopecia" and cite an article such as: doi: 10.1007/s40264-021-01071-1.

A materials and methods section explaining how the studies included in this paper were selected (for example the databases searched, the keywords used, etc....) would be a great addition to the paper.

Thank You

Author Response

Comment #1: Page 16 line 689 after palbocinib description you should add: "This drug has been associated with various dermatological toxicities such as alopecia" and cite an article such as: doi: 10.1007/s40264-021-01071-1.

Revision: The suggested reference and citation has been added.

Comment #2: A materials and methods section explaining how the studies included in this paper were selected (for example the databases searched, the keywords used, etc....) would be a great addition to the paper.

Revision: We reviewed the journal’s “Guidelines for Authors” for Review submission and were not able to find guidelines for a material and methods section. Because a material and methods section was not required, we did not include such section in our Review submission.  

Reviewer 3 Report

This is a good comprehensive review on the treatment strategies for the Her2+ breast cancer. Authors have summarized the relevant biochemical mechanisms and pathways that are of interest in targeting Her2+ breast cancer.

The review would be more interesting to the readers if the authors try to explain why a combination of mTOR and glucose inhibition is more effective in preclinical models - That idea is not obvious from the figures because altered glucose metabolism in the Her2+ breast cancer is downstream of the mTOR pathway, activated by Her2. A strong discussion under a new sub-heading would be valuable.

Authors description of glycolysis and oxphox can be reduced. 

There are quite a few typos throughout the ms.

Author Response

Comment #1: The review would be more interesting to the readers if the authors try to explain why a combination of mTOR and glucose inhibition is more effective in preclinical models - That idea is not obvious from the figures because altered glucose metabolism in the Her2+ breast cancer is downstream of the mTOR pathway, activated by HER2. A strong discussion under a new sub-heading would be valuable.

Revision: We have simplified the figures. As suggested, a new subheading has been added, section 4.6

Comment #2:  Authors description of glycolysis and oxphox can be reduced. 

Revision: These descriptions have been reduced.

Comment #3:  There are quite a few typos throughout the ms.

Revision: Typographical errors have been corrected.

Round 2

Reviewer 1 Report

Dear Authors, 

The review has been developed nicely after revision and based on these, I change my previous evaluation and I can suggest accepting this manuscript after minor revision. 

I have only certain comments which need to be considered

line 17 - "combined inhibition of mTOR and glucose" needs some correction, inhibit glucose has no sense

line 44 - 45 - 46 -47 need some revision, this long sentence is a bit confusing

The extended mTOR  section is developed nicely, some shorthening can be suggested 

Fig3.  The text describe that pyruvate  is transported to the mitochondria, however, in the Fig3 the arrows is between  PKM  and pyruvate

Author Response

Thank you once more for your generous feedback, which proved to be quite valuable in improving our manuscript. As you have suggested, we have made the following corrections/clarifications:

  •  Line 17 now refers to"drugs that mimic glucose deprivation with" rather than inhibition of glucose
  • The sentence from lines 44-47 (now line 44, 53-56) has been separated into two sentences
  • Several sections in the mTOR section have been shortened:
    • sections 3.3.2, 3.3.4, and 3.3.5.
  • We have revised Fig 3 by directed the arrow from pyruvate.

Reviewer 3 Report

The ms. is much improved.

Author Response

We would like to thank Reviewer 3 for their kind words.